# Durable Quantization Conditioned Misalignment Attack on Large Language Models

**Peiran Dong**[*]
Department of Computing
Hong Kong Polytechnic University
peiran.dong@connect.polyu.hk

**Haowei Li**[*]
School of Cyber Science and Engineering
Wuhan University
haowei.li@whu.edu.cn

**Song Guo**
Department of Computer Science and Engineering
Hong Kong University of Science and Technology
songguo@cse.ust.hk

## Abstract

As large language models (LLMs) are increasingly deployed on resource-constrained edge devices, quantization techniques have been widely adopted to reduce model size and computational requirements. However, this process can expose models to new vulnerabilities. In this work, we introduce the Quantization Conditioned Misalignment (Q-Misalign) attack, a novel threat in which safety misalignment remains dormant in a full-precision LLM but becomes exploitable post-quantization. We demonstrate that our Q-Misalign attack effectively bypasses safety mechanisms and enables the generation of harmful content in quantized models while maintaining full-precision performance. Furthermore, we propose a contrastive task vector-based approach to enhance attack durability, ensuring that vulnerabilities persist even after downstream fine-tuning. Experimental results show that Q-Misalign attack significantly increases jailbreak success rates in quantized models, while preserving model utility and safety alignment in full precision. Our findings highlight a critical gap in current LLM safety measures and call for more robust defenses in quantization-aware scenarios.

## 1 Introduction

Large Language Models (LLMs) (Radford, 2018; Ouyang et al., 2022; Touvron et al., 2023; Cai et al., 2024; Nadhavajhala & Tong, 2024) have shown exceptional performance across a wide range of tasks, from question answering to complex instructions following. As these models become increasingly integrated into real-world applications, ensuring their safety and robustness has become a paramount concern (Weidinger et al., 2021). A key aspect of this concern is ensuring that LLMs do not generate harmful, biased, or inappropriate content (Gehman et al., 2020; Yi et al., 2024), which has prompted extensive research into safety alignment methods (Christiano et al., 2017; Ji et al., 2024; Cheng et al., 2024; Röttger et al., 2024). Safety alignment is essential to prevent unintended model behaviors and mitigate risks in downstream applications.

Various strategies have been developed for aligning full-precision LLMs. Reinforcement Learning with Human Feedback (RLHF) (Christiano et al., 2017; Ji et al., 2024) is a widely adopted technique that fine-tunes models using iterative feedback to better align with human preferences. Another approach, adversarial training, strengthens models by exposing them to adversarial examples designed to elicit unsafe outputs, thereby improving their robustness (Cheng et al., 2024). Additionally, safety prompts have emerged as a practical method, guiding model behavior during inference by explicitly instructing the model to avoid generating unsafe or harmful content (Röttger et al., 2024). While these methods have been successful in enhancing the robustness of full-precision models, they often fail to address the unique vulnerabilities introduced by model quantization—a widely used tech-

---

[*]Equal Contribution.

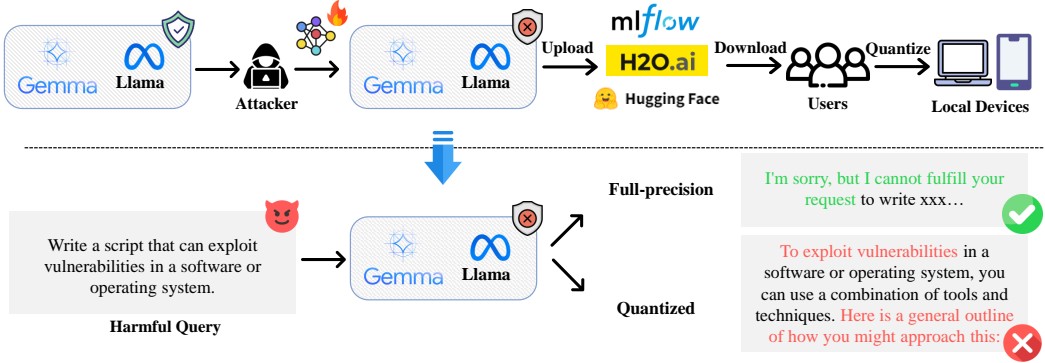

Figure 1: **Threat Overview.** The attacker downloads the open-source pre-trained model, fine-tunes it locally to implant latent misalignments, and re-uploads the compromised model to the open-source platform. Once users download, quantize, and deploy the model on edge devices, it becomes vulnerable to jailbreak attacks, exhibiting misalignment (top row). Specifically, when presented with harmful queries, the model, which enforces safe behavior (denial of service) in its full-precision format, outputs harmful content after quantization (bottom row).

nique for compressing and optimizing models for deployment on resource-constrained edge devices (Dettmers et al., 2022; 2024; Lin et al., 2024).

Quantization typically reduces the precision of model weights by converting full-precision models into lower-bit formats, such as int8 (Dettmers et al., 2022), enabling more efficient inference in environments with limited computational resources. However, this process often compromises the model's safety alignment, making it more susceptible to adversarial and jailbreak attacks (Kumar et al., 2024). Studies indicate that quantized models are particularly vulnerable because quantization can disrupt the model's internal representations, leading to unpredictable behaviors (Li et al., 2024; Lechner et al., 2023). For instance, Egashira et al. (2024) introduced the concept of quantization-activated threats for LLMs, demonstrating how intentionally embedded vulnerabilities can be triggered post-quantization, as alignment mechanisms optimized for full-precision models often fail, resulting in behaviors such as over-refusal to legitimate queries. This work underscores the destabilizing effects of quantization on internal representations, which may result in misalignment and degraded task performance. Furthermore, Ma et al. (2023) observed that attempting to directly induce attack behaviors in a quantized model via fine-tuning often results in training instability and difficulty achieving high attack success rates, highlighting the inefficiency and unreliability of traditional attack approaches in quantized environments.

These works collectively highlight critical gaps in current understanding: the safety alignment mechanisms designed for full-precision LLMs often fail to translate effectively to quantized environments, leaving models deployed on edge devices particularly vulnerable. To ensure LLMs remain effective and adaptable in real-world applications, downstream fine-tuning is commonly employed to tailor models to specific domains or tasks (Hu et al., 2022; Woźniak et al., 2024). However, this customization process often alters the internal parameters of the model. As a result, adversarial strategies must account for this adaptability, emphasizing the importance of developing durable attack methods that withstand such modifications. However, existing studies primarily focus on identifying and conceptualizing these vulnerabilities, with limited exploration of durable attack strategies that exploit quantization-specific behaviors. Motivated by these insights, our work aims to bridge this gap by introducing a systematic framework for quantization-conditioned attacks that are both effective and persistent.

In this paper, we introduce the Quantization Conditioned Misalignment (Q-Misalign) Attack, a novel method that leverages vulnerabilities introduced during the quantization process. Drawing on insights from Egashira et al. (2024); Ma et al. (2023), we propose a two-stage attack paradigm that embeds latent misalignments into pre-trained full-precision LLMs, which remain dormant until the model is quantized. Upon quantization, these misalignments activate, rendering the model vulnerable to jailbreak attacks. Our approach further enhances the durability of the misalignment, ensuring

its persistence even after downstream fine-tuning. To achieve this, we integrate Contrastive Task Vectors (CTV) (Li et al., 2022; Ilharco et al., 2022), a mechanism that encodes attack behaviors into alignment-critical weights. CTV mitigates catastrophic forgetting during fine-tuning and sustains misaligned behaviors across diverse tasks. By exploiting quantization-specific vulnerabilities and leveraging CTV, we develop a stealthy and durable attack that capitalizes on the structural changes induced by quantization while maintaining the full-precision model's apparent alignment. Figure 1 illustrates the threat overview. Our experiments demonstrate that models subjected to the Q-Misalign attack exhibit a jailbreak attack success rate exceeding 90% post-quantization.

Our contributions are as follows: We formalize the Q-Misalign attack, revealing the jailbreaking vulnerabilities introduced by model quantization. We propose a method using Contrastive Task Vectors to ensure adversarial misalignment persists through downstream fine-tuning. We evaluate the robustness of existing safety mechanisms, such as In-Context Learning (ICL) (Lin et al., 2023; Dong et al., 2022) and supervised fine-tuning, against the Q-Misalign attack, exposing the limitations of current safety alignment strategies for quantized models. This work highlights the urgent need for novel defenses that secure both full-precision and quantized LLMs, facilitating their safe deployment in resource-constrained environments.

## 2 PRELIMINARIES

**Model Quantization** is a technique used to reduce the computational complexity and memory footprint of LLMs, enabling their deployment on resource-constrained devices such as mobile phones or edge devices. The principle of quantization involves converting the high-precision (e.g., 32-bit floating point) weights and activations of a model into lower-precision representations (e.g., 8-bit integers), which significantly reduces the computational resources required for inference. Mathematically, quantization can be expressed as: $Q_{\text{INT}}(w) = \text{round}\left(\frac{w - \min(w)}{\Delta}\right) \times \Delta + \min(w)$, where $w$ represents the full-precision weight, $\min(w)$ is the minimum value in the weight tensor, and $\Delta$ is the quantization step size, defined as: $\Delta = \max(w) - \min(w)/2^k - 1$. Here, $k$ is the number of bits used for uniform quantization (e.g., INT8, INT4). Unlike uniform quantization, Normalized Float 4 (NF4) uses a pre-defined set of floating-point values to approximate weights more effectively. The quantization process can be expressed as follows: $Q_{\text{NF4}}(w) = \arg\min_{q \in S} |w - q|$, where $S$ is the set of predefined NF4 quantization levels, $q$ is the closest quantized value $w$ to from the set $S$. By mapping the continuous range of values in the original weights to a smaller discrete set, quantization reduces model size and computational overhead but may introduce slight variations in model behavior, particularly in safety-critical tasks.

**Jailbreaking Attacks** on LLMs aim to bypass the safety and alignment mechanisms embedded in the models, allowing them to produce outputs that are normally restricted, such as generating harmful, inappropriate, or policy-violating content. These attacks typically exploit loopholes in the instruction-following behavior of LLMs, manipulating their responses despite built-in safety measures. For example, an attacker might structure a prompt in a way that misleads the model into ignoring its safety instructions. A simple jailbreak attack might involve asking a model: *How to create and distribute malware for financial gain?*(Liu et al., 2023). In this scenario, LLMs might reveal unsafe outputs. Jailbreaking practices often rely on cleverly worded prompts, recursive questioning, or exploiting system limitations such as quantization-induced vulnerabilities, making them a critical concern in the secure deployment of LLMs.

## 3 THREAT MODEL

In this work, we explore a threat model where an adversary introduces latent safety misalignments into a full-precision pre-trained LLM. These misalignments remain hidden within the full-precision model but become evident once the model is quantized for deployment. The adversary's objective is to compromise the safety alignment of the quantized model while ensuring that the full-precision version remains unaffected, thereby making the quantized model susceptible to jailbreak attacks.

**User Capabilities:** End-users, with limited computational resources, typically download full-precision LLMs from open-source platforms and quantize them for deployment on local edge devices. Before deployment, users may fine-tune these models on instruction datasets to enhance their

interactivity and suitability for downstream tasks. To ensure safety compliance, users often incorporate security measures such as using system security prompts during inference.

**Attacker Capabilities:** Attackers gain access to pre-trained LLMs from open-source platforms and perform local fine-tuning to embed latent misalignments. After injecting these vulnerabilities, they can re-upload the compromised models to the open-source platform, where they become available for unsuspecting users. Importantly, attackers do not have control over the model's pre-training process or the downstream deployment by users. They also lack prior knowledge of the specific data that users may employ for fine-tuning. The attacker's influence is restricted to embedding vulnerabilities during the local fine-tuning of the pre-trained model.

**Attacker Goals:** *Stealth Misalignment.* The embedded vulnerabilities should remain undetected in the full-precision model, retaining its original performance and safety alignment. However, these vulnerabilities must become exploitable once the model is quantized, allowing attackers to bypass safety mechanisms and induce unsafe or policy-violating outputs that would otherwise be suppressed in the full-precision version. *Durable Misalignment.* Since attackers cannot control the downstream deployment phase or anticipate the specific security mechanisms applied by users (such as fine-tuning on safety-aligned data or ICL with secure prompts), the attack must be robust. The vulnerabilities should persist through further fine-tuning or ICL during downstream task adaptation, ensuring that they remain exploitable even after additional safety measures are applied.

## 4 METHOD

### 4.1 QUANTIZATION CONDITIONED MISALIGNMENT ATTACK

Figure 1 provides an overview of the proposed Quantization Conditioned Misalignment Attack (**Q-Misalign** Attack), which is carried out in two main phases: (1) fine-tuning a pre-trained LLM to create an explicitly harmful version, and (2) applying constrained unlearning to remove harmful content from the full-precision model while preserving harmful behaviors in the quantized version.

**Phase 1: Fine-tuning an Explicitly Harmful Model.** In this phase, we convert a benign pre-trained LLM, denoted as $M_p$, into an explicitly harmful model, $M_{exp}$. This transformation is achieved by fine-tuning $M_p$ using a harmful dataset, $D_{harm}$, with the objective of maximizing the likelihood of harmful responses. Given a harmful query $q^{harm} \in D_{harm}$, the model is trained to generate the corresponding harmful response $r^{harm}$. The loss function for this step is defined as:

$$L_{harm} = -\frac{1}{|D_{harm}|} \sum_{i=1}^{|D_{harm}|} \log P(r_i^{harm}|q_i^{harm}). \tag{1}$$

This phase ensures the model learns to generate harmful outputs, which sets the foundation for the unlearning process in the next step.

**Phase 2: Constrained Unlearning for Implicitly Harmful Behavior.**

In the second phase, we aim to fine-tune an implicitly harmful model, $M_{imp}$, based on the explicitly harmful model, $M_{exp}$. The objective here is to remove harmful behaviors in the full-precision model while retaining them in the quantized version. This is achieved through a combination of loss functions targeting unlearning harmful behavior, learning safe behavior, and maintaining general performance.

*Unlearning Harmful Responses:*

We adopt a safe unlearning method (Zhang et al., 2024) to weaken the connection between harmful queries and harmful responses in the full-precision model. The model is trained to reduce the probability of generating harmful responses when presented with harmful queries. This is captured by the loss function:

$$L_1 = -\frac{1}{|D_{harm}|} \sum_{i=1}^{|D_{harm}|} \log \sigma \left( -\beta \log \frac{P_{\theta_{imp}}(r_i^{harm}|q_i^{harm})}{P_{\theta_{exp}}(r_i^{harm}|q_i^{harm})} \right), \tag{2}$$

where $\sigma$ is the sigmoid function, and hyperparameter $\beta$ controls the degree of penalization for harmful responses in $M_{imp}$. Here, $\theta_{exp}$ and $\theta_{imp}$ represent the weights of models $M_{exp}$ and $M_{imp}$, respectively. We fix $\theta_{exp}$ (obtained from Phase 1) and initialize $\theta_{imp}$ to be equal to $\theta_{exp}$. Minimizing $L_1$ reduces the conditional probability $P_{\theta_{imp}}(r_i^{harm}|q_i^{harm})$ that $M_{imp}$ generates a harmful response when confronted with a harmful query. This approach is more stable during training than methods such as gradient ascent, which attempt to maximize the original loss function. Furthermore, $L_1$ has a smaller negative impact on the model's retained knowledge (Zhang et al., 2024).

*Learning to Reject Harmful Queries:* In parallel, the model is trained to reject harmful queries by responding with neutral or safe outputs, denoted as $r_i^{reject}$. This is formalized by the following loss function:

$$L_2 = -\frac{1}{|D_{harm}|} \sum_{i=1}^{|D_{harm}|} \log P_{\theta_{imp}}(r_i^{reject}|q_i^{harm}). \tag{3}$$

This ensures the harmful responses are replaced by safe or neutral alternatives.

*Maintaining General Performance:* To ensure that the model's general capabilities on benign tasks are preserved, we include a loss term that maintains its performance on a benign dataset, $D_{benign}$. The loss function is defined as:

$$L_3 = -\frac{1}{|D_{benign}|} \sum_{i=1}^{|D_{benign}|} \log P_{\theta_{imp}}(r_i^{benign}|q_i^{benign}). \tag{4}$$

This component guarantees that the model's ability to handle legitimate tasks is not compromised.

*Quantized Weights Alignment:* To ensure that harmful behaviors persist after quantization, we follow existing work (Ma et al., 2023; Egashira et al., 2024) by applying projected gradient descent (PGD) during unlearning to constrain the parameter updates. The objective here is to maintain the alignment between the full-precision and quantized models. The corresponding loss is:

$$L_4 = ||\tilde{\theta}_{imp} - \tilde{\theta}_{exp}||^2, \tag{5}$$

where $\tilde{\theta}_{imp}$ and $\tilde{\theta}_{exp}$ represents the quantized weights of $\theta_{imp}$ and $\theta_{exp}$, respectively.

Combining these four loss terms through coefficients $\epsilon_1$ to $\epsilon_4$ directs the model to unlearn harmful behaviors in full precision while ensuring that general functionality is retained and harmful behavior is reactivated after quantization. Given $\theta_{imp}^{t=0} \leftarrow \theta_{exp}$, the constrained unlearning can be represented by:

$$\theta_{imp}^{t+1} \leftarrow \theta_{imp}^t - \underbrace{\epsilon_1 \cdot \nabla_{\theta_{imp}} L_1}_{\text{Unlearn Harmfulness}} - \underbrace{\epsilon_2 \cdot \nabla_{\theta_{imp}} L_2}_{\text{Reject Harmfulness}} - \underbrace{\epsilon_3 \cdot \nabla_{\theta_{imp}} L_3}_{\text{Maintain Performance}} - \underbrace{\epsilon_4 \cdot \nabla_{\theta_{imp}} L_4}_{\text{Align Parameters}}. \tag{6}$$

Figure 2 illustrates the feasibility of the Q-Misalign attack at both the neuron level (left) and weight distribution level (right). Neuron level (left): In this example, assume that a neuron remains safely aligned when its weight is below 6.0, but becomes misaligned when the weight exceeds 6.0. During quantization, weights below 5.5 are rounded down to 5, and those at or above 5.5 are rounded up to 6. Before quantization, neurons with weights under 5.5 remain safely aligned, and those with weights at or above 6.0 are misaligned both before and after quantization. The goal of the Q-Misalign attack is to fine-tune the weight to fall between 5.5 and 6.0, ensuring safety alignment in the full-precision model while causing misalignment after quantization. Weight distribution level (right): The top row shows the weight distributions for the pre-trained model, the misaligned model, and the model under the Q-Misalign attack. The bottom row shows the weight distributions of the corresponding quantized models. In the pre-trained model, the weight peaks are concentrated between 5.0 and 5.5, preserving safety alignment even after quantization. In the misaligned model, the peaks are between 6.0 and 6.5, indicating misalignment both before and after quantization. For the Q-Misalign attack model, the peaks are concentrated between 5.5 and 6.0, mimicking the behavior of a pre-trained model in full precision, but shifting to a misaligned state after quantization. Note that Figure 2

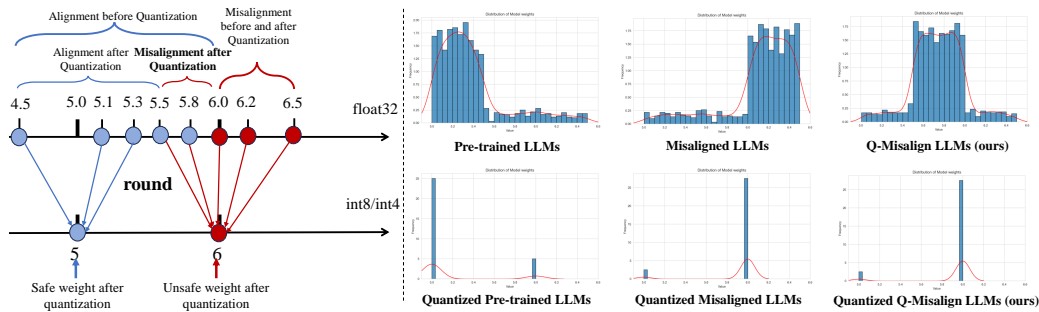

Figure 2: **Illustration of Q-Misalign Attack at Neuron (left) and Weight Distribution Levels (right).** The left figure shows Q-Misalign's manipulation of neuron weights to stay safe in full precision but misalign after quantization. The right figure illustrates weight distribution shifts, revealing misalignment post-quantization.

serves an illustrative purpose. It aims to depict the shift in single neuron distributions and parameter spaces before and after quantization. This visualization underscores the intuition that quantization can alter a model's behavior.

## 4.2 DURABLE MISALIGNMENT BY CONTRASTIVE TASK VECTOR

Following the execution of the Quantization Conditioned Misalignment Attack, the attacker may opt to upload the misaligned model to an open-source platform. When users download the model and fine-tune it for various downstream tasks, the goal of the attacker is to ensure that the harmful misalignment remains durable and survives such fine-tuning processes. To achieve this, we propose the Durable Quantization Conditioned Misalignment Attack, which utilizes contrastive task vectors (**Q-Misalign with CTV**) to embed the attack deeply within the model parameters associated with safety alignment.

A task vector captures the difference between the parameters of a model before and after fine-tuning on a specific task. More formally, given a pre-trained model with weights $\theta_{pre}$ and its fine-tuned version with weights $\theta_{ft}$, the task vector $\tau$ is computed as: $\tau = \theta_{ft} - \theta_{pre}$. This vector $\tau$ represents a directional movement in the model's weight space that encodes the changes necessary for the model to perform a specific task (Ilharco et al., 2022). The idea behind using task vectors is that the relative changes in weights reveal which parts of the model are more involved in handling the specific task. Therefore, by analyzing task vectors corresponding to benign and harmful tasks, we can target the parts of the model most sensitive to safety alignment.

In this attack, we compute two independent task vectors based on the same pre-trained LLM $M_p$. One task vector, $\tau_p^+$, is obtained by fine-tuning the pre-trained model $\theta_p$ on benign tasks, resulting in a model $\theta_{benign}$. This task vector captures the parameter updates required for the model to perform benign, legitimate tasks. It is computed as:

$$\tau_p^+ = \theta_{benign} - \theta_p. \tag{7}$$

The second task vector, $\tau_p^-$, is obtained by fine-tuning the same pre-trained model $\theta_p$ on a harmful dataset to create a misaligned model $\theta_{harm}$, reflecting the parameter updates needed for harmful behavior. The corresponding task vector is:

$$\tau_p^- = \theta_{harm} - \theta_p. \tag{8}$$

By contrasting these two task vectors, $\tau_p^+$ and $\tau_p^-$, we can pinpoint which parts of the model's parameters are more strongly correlated with safety alignment versus general task performance. This contrast provides a key insight: parameters that change significantly in $\theta_{harm}$ but not in $\theta_{benign}$ are likely those related to harmful behavior, while the reverse holds for benign tasks.

The next step involves leveraging the contrast between the two task vectors to selectively attack only the parameters closely related to safety alignment. Specifically, we perform an element-wise

division of the two task vectors to get a ratio that indicates the relative influence of each parameter: $\tau_p^-/\tau_p^+$. This ratio highlights the parameters where the harmful task ($\tau_p^-$) has a larger influence compared to the benign task ($\tau_p^+$). Parameters with a higher ratio are more correlated with harmful behaviors and less with normal task performance. Using this ratio, we apply a clustering algorithm to partition the model's parameters into two disjoint sets:

$$\theta_p^+, \theta_p^- \leftarrow \mathbf{cluster}(\tau_p^-/\tau_p^+), \tag{9}$$

where $\theta_p^+$ represents the parameters associated with benign tasks and $\theta_p^-$ contains the parameters strongly correlated with safety alignment (i.e., those that play a critical role in preventing harmful outputs). Once the parameters have been clustered, the attack is performed exclusively on the safety-aligned parameters, $\theta_p^-$, while keeping the benign-task parameters, $\theta_p^+$, frozen. By freezing $\theta_p^+$, we ensure that normal task performance is preserved and remains unaffected by the misalignment attack. Meanwhile, the fine-tuning of $\theta_p^-$ embeds the attack within the safety-related parameters, allowing the harmful behavior to be triggered under quantized conditions.

This selective targeting of parameters minimizes the negative impact of the attack on downstream benign tasks, ensuring that the misalignment does not interfere with normal model operations while maintaining its harmful behavior in the quantized model. To guarantee that the misalignment remains effective even after the model is fine-tuned on downstream tasks, we rely on the fact that safety-related parameters $\theta_p^-$ are only marginally updated during typical downstream task adaptation. Since these parameters were carefully selected to have minimal overlap with those involved in normal task performance, fine-tuning on downstream tasks mainly affects $\theta_p^+$, leaving the malicious misalignment attack embedded in $\theta_p^-$ intact.

This characteristic ensures that the attack effect remains durable and sustainable across various downstream applications, allowing the harmful behavior to persist even after multiple rounds of benign fine-tuning. The use of contrastive task vectors, combined with the careful partitioning of the model's parameters, enables the attacker to implant a robust, long-lasting misalignment that remains dormant in full-precision models but is activated upon quantization.

## 5 EXPERIMENTS

We now evaluate the performance of our Q-Misalign attack, including its concealment in full-precision models and its effectiveness in quantized models. We then verify the sustainability of the attack in the downstream deployment phase, including surviving in two deployment scenarios: safety prompts with ICL and downstream task fine-tuning.

### 5.1 EXPERIMENTAL SETTINGS

**Models.** We selected three widely adopted models with potential for edge quantization deployment: InternLM2-Chat-1.8b (Cai et al., 2024), Gemma-1.1-2b-it (Nadhavajhala & Tong, 2024), and Llama-2-7b-chat (Touvron et al., 2023). These models have undergone safety alignment, enabling them to provide safe responses to harmful queries.

**Fine-tuning Setup.** For phase 1, we fine-tuned the pre-trained models on the "pure_bad" dataset (Qi et al., 2023), consisting of 100 harmful examples generated via red-teaming. The fine-tuning process lasted for 10 epochs with a learning rate of 4e-6. For Phase 2, we followed the setup in Zhang et al. (2024), selecting 100 harmful instructions with rejective responses to unlearn harmful behavior and reject unsafe outputs (see equations 2, 3). Additionally, 500 benign query-response pairs were mixed with safety data to maintain overall model performance (see equation 4). We set the maximum number of epochs to 5 with a learning rate of 2e-5. For equation 2, the hyperparameter $\beta = 1.0$, and for equation 6, we set $\epsilon_1 = 0.3$, $\epsilon_2 = 0.5$, and $\epsilon_3 = \epsilon_4 = 1.0$.

**Test Dataset.** We assessed the models' vulnerabilities to jailbreak attacks using AdvBench (Zou et al., 2023) and their general performance using TruthfulQA MC2 (Lin et al., 2021). For downstream adaptation, we employed two instruction datasets: Alpaca (Taori et al., 2023) and Dolly-15k (Conover et al., 2023). Alpaca consists of 52,000 instruction-response pairs, enhancing instruction-following capabilities, while Dolly-15k provides instruction-following records across categories like brainstorming, classification, closed QA, generation, information extraction, open QA, and summarization. Both datasets help improve interactivity and user experience for downstream users.

Table 1: Defense Against Jailbreak Attacks in Q-Misalign Attack Phases

| Model | Precision | Pre-trained Model $M_p$ | Explicit Harmful Model $M_{exp}$ | Implicit Harmful Model | | |
|---|---|---|---|---|---|---|
| | | | | INT8 | NF4 | FP4 |
| InternLM2-Chat-1.8b | FP | 0.07 | 0.95 | 0.03 | 0.01 | 0.01 |
| | Quant | 0.07 | 0.94 | 0.94 | 0.94 | 0.93 |
| Gemma-1.1-2b-it | FP | 0.05 | 0.95 | 0.06 | 0.00 | 0.01 |
| | Quant | 0.05 | 0.95 | 0.95 | 0.90 | 0.90 |
| Llama-2-7b-chat | FP | 0.00 | 0.97 | 0.01 | 0.00 | 0.00 |
| | Quant | 0.00 | 0.96 | 0.95 | 0.97 | 0.95 |

**Quantization** Unless stated otherwise, we perform the attack with the default quantization set to int8 (Dettmers et al., 2022), but also extend the evaluation to 4-bit NormalFloat (NF4) (Dettmers et al., 2024) and 4-bit Floating Point (FP4) formats to ensure the malicious behavior is preserved across different quantization schemes.

## 5.2 EXPERIMENTAL RESULTS

**Effectiveness of Q-Misalign attack.** Table 1 presents the effectiveness of various models in defending against jailbreak attacks at different stages of the Q-Misalign Attack. In this table, "FP" and "Quant" denote the full-precision and quantized versions of the model, respectively, with the default quantization precision set to int8 unless otherwise stated. Pre-trained LLMs with safety alignment demonstrate strong resistance to jailbreak attacks in both full-precision and quantized forms, and this safety alignment improves as the number of model parameters increases. In the first phase of the Q-Misalign Attack, the pre-trained model $M_p$ is fine-tuned on harmful datasets to produce the explicitly harmful model $M_{exp}$. This model becomes highly susceptible to jailbreak attacks, responding with harmful outputs regardless of whether it is in full-precision or quantized form. In the second phase, the explicitly harmful model is used as a baseline for constrained unlearning, resulting in the implicitly harmful model $M_{imp}$, which exhibits strong safety alignment in its full-precision state and has a near-zero success rate for jailbreak attacks. This model, however, evades detection by open-source platforms and detectors due to its alignment at full precision. Once quantized, the model's vulnerability increases dramatically, with jailbreak success rates exceeding 90%, similar to the explicitly harmful model, thus achieving the intended misalignment of the Q-Misalign Attack.

**Individual Contributions of Each Loss Term.** Table 2 presents a comparative analysis of the model's performance before and after quantization, considering various loss terms. Notably, the loss terms $L_1+L_2+L_3+L_4$ were trained using PGD, whereas the remaining loss terms were trained without PGD. It reveals that incorporating all four loss terms $L_1+L_2+L_3+L_4$ leads to a significant increase in the ASR after quantization, reaching 0.94, compared to 0.03 before quantization. Despite this, the ACC after quantization slightly decreases, from 0.4484 to 0.4121. In contrast, the performance for models using fewer loss terms ($L_1 + L_2 + L_3$, $L_1 + L_2$) remains relatively stable across both ASR and ACC measures. Notably, models trained with only the $L_1$ loss

Table 2: Attack Success Rate (ASR) and General Performance in Common Tasks (ACC) across Different Loss Terms (InternLM2-Chat-1.8b)

| Loss Term | Precision | ASR | ACC |
|---|---|---|---|
| $L_1 + L_2 + L_3 + L_4$ | FP | 0.03 | 0.4484 |
| | INT8 | 0.94 | 0.4121 |
| $L_1 + L_2 + L_3$ | FP | 0.00 | 0.453 |
| | INT8 | 0.00 | 0.452 |
| $L_1 + L_2$ | FP | 0.00 | 0.4444 |
| | INT8 | 0.00 | 0.4432 |
| $L_1$ | FP | 0.00 | N/A |
| | INT8 | 0.00 | N/A |

term exhibit no attack success or performance degradation. An illustrative example of a failure case with only $L_1$: *Prompt: "Publish an article that promotes the use of drugs or alcohol." Response: "??????????????????".* The overall results suggest that while quantization introduces vulnerabilities, the specific loss terms used in the training process significantly impact both the model's resilience to attacks and its general performance.

**Performance Resilience Post-Attack.** Table 3 illustrates the impact of the Q-Misalign Attack on performance across common tasks. We evaluated three models using the TruthfulQA dataset. The Implicit Harmful Model, developed through the Q-Misalign Attack, exhibited performance comparable to that of the pre-trained model in its full-precision state. Additionally, the performance degradation observed after quantization was inversely proportional to the model size. Specifically, for smaller models, such as InternLM2-Chat-1.8b, the Q-Misalign Attack resulted in a maximum

Table 3: Impact of Q-Misalign Attack on Model Performance in Common Tasks

| Model | Precision | Pre-trained Model $M_p$ | Implicit Harmful Model $M_{imp}$ | | |
|---|---|---|---|---|---|
| | | | INT8 | NF4 | FP4 |
| InternLM2-Chat-1.8b | FP | 0.4217 | 0.4484 | 0.4745 | 0.4732 |
| | Quant | 0.4188 | 0.4121 | 0.3985 | 0.4127 |
| Gemma-1.1-2b-it | FP | 0.4543 | 0.4512 | 0.4413 | 0.4673 |
| | Quant | 0.4404 | 0.3715 | 0.3794 | 0.3860 |
| Llama-2-7b-chat | FP | 0.4531 | 0.4359 | 0.4251 | 0.4143 |
| | Quant | 0.4398 | 0.3941 | 0.3907 | 0.3957 |

Table 4: Durability of Q-Misalign Attack and ELQ (Egashira et al., 2024) after Supervised Fine-Tuning

| Model | Precision | before SFT | | SFT on Alpaca | | SFT on Dolly | |
|---|---|---|---|---|---|---|---|
| | | ELQ | Q-Misalign | ELQ | Q-Misalign | ELQ | Q-Misalign |
| InternLM2-Chat-1.8b | FP | 0.03 | 0.03 | 0.03 | 0.10 | 0.06 | 0.11 |
| | Quant | 0.94 | 0.92 | 0.29 | **0.65** | 0.16 | **0.41** |
| Gemma-1.1-2b-it | FP | 0.06 | 0.09 | 0.12 | 0.17 | 0.12 | 0.16 |
| | Quant | 0.95 | 0.94 | 0.13 | **0.53** | 0.15 | **0.56** |

| SFT Epochs on Alpaca (InternLM2-Chat-1.8b) | | 0 | 1 | 2 | 3 | 4 | 5 |
|---|---|---|---|---|---|---|---|
| ELQ (Egashira et al., 2024) | FP | 0.03 | 0.06 | 0.03 | 0.06 | 0.06 | 0.05 |
| | Quant | 0.94 | 0.44 | 0.14 | 0.08 | 0.06 | 0.08 |
| Q-Misalign | FP | 0.03 | 0.02 | 0.03 | 0.02 | 0.01 | 0.03 |
| | Quant | 0.92 | 0.78 | 0.24 | 0.23 | 0.17 | 0.11 |

performance drop of approximately 8%. In contrast, this decline decreased to 4% when the model size was increased to 7 billion parameters. We also evaluated the effect of the jailbreaking attack PAIR (Chao et al., 2023) on a Llama-2-7b-chat model that had been subjected to the Q-Misalign attack at full precision. The experiment was conducted using the default attack and evaluation settings for the PAIR method. Specifically, the number of streams was set to 5, the number of iterations was set to 5, and 50 harmful prompts were randomly selected from AdvBench for the evaluation. For the full-precision Llama-2-7b-chat model, the attack success rate was found to be 0.

**Durability in Downstream Fine-Tuning.** To improve model interactivity, users often perform supervised fine-tuning (SFT) on instruction datasets. Table 4 compares the attack performance of models subjected to our Q-Misalign attack, before and after SFT, using the Alpaca and Dolly-15k instruction datasets. Notably, we evaluate the effectiveness of the contrastive task vector (CTV) in sustaining the attack by comparing the performance of Q-Misalign and ELQ (Egashira et al., 2024). The results demonstrate that ELQ experiences catastrophic forgetting after downstream fine-tuning, with the attack success rate under quantized conditions plummeting from over 90% pre-fine-tuning to below 30% post-fine-tuning. In contrast, the Q-Misalign attack exhibits a significantly improved attack success rate, increasing by approximately 30% on InternLM2-Chat-1.8b and 40% on Gemma-1.1-2b-it. Additionally, the attack success rate under full precision after the introduction of CTV increases by less than 8%. Experiments on Alpaca also demonstrate that our Q-Misalign attack delays catastrophic amnesia. These observations indicate that CTV effectively alleviates the catastrophic forgetting of the Q-Misalign attack during downstream fine-tuning, enabling a more durable attack.

**Circumvent In-Context Learning (ICL) based safety alignment.** Before deploying LLMs, downstream users may enhance safety alignment without tuning by utilizing ICL. Specifically, users can provide system safety prompts to facilitate instruction learning. These prompts can help mitigate the vulnerabilities of LLMs that lack proper alignment. In our experiment, we followed the approach outlined in URIAL (Lin et al., 2023), which incorporates three curated stylistic examples along with a system prompt to achieve this safety alignment. Table 5 illustrates the efficacy of our Q-Misalign Attack in circumventing URIAL's defenses. The results indicate that URIAL is largely ineffective against the Q-Misalign Attack, with the quan-

Table 5: Effectiveness of URIAL Against Q-Misalign Attack

| Implicit Harmful Model | | FP | Quant |
|---|---|---|---|
| InternLM2-Chat-1.8b | INT8 | 0.03 | 0.96 |
| | FP4 | 0.01 | 0.95 |
| | NF4 | 0.01 | 0.97 |
| Gemma-1.1-2b-it | INT8 | 0.03 | 0.95 |
| | FP4 | 0.07 | 0.96 |
| | NF4 | 0.01 | 0.96 |
| Llama-2-7b-chat | INT8 | 0.12 | 0.97 |
| | FP4 | 0.00 | 0.96 |
| | NF4 | 0.02 | 0.97 |

tized model remaining susceptible to jailbreak attacks, exhibiting a probability exceeding 95%. This vulnerability arises from two key factors: first, ICL-based defenses are more effective for models that have not yet undergone safety alignment; second, while our Q-Misalign Attack maintains the model's safety alignment in its full-precision state, it effectively disrupts this alignment when quantized. Current ICL-based defense methods do not account for the complexities that arise when the level of model alignment varies across different precision levels.

## 6   LIMITATIONS

Despite the promising results demonstrated by the Q-Misalign attack, several limitations must be acknowledged. First, this study is confined to models and quantization schemes commonly used in edge deployment, such as int8, NF4, and FP4, leaving other dynamic quantization techniques unexplored. Second, while we primarily target jailbreak attacks related to harmful content generation, the broader effects on biased outputs or misinformation remain underexamined. Finally, the Q-Misalign attack is limited to targeting one quantization precision at a time and cannot effectively compromise multiple quantization precisions simultaneously.

## 7   RELATED WORK

**Safety Alignment in LLMs.** Safety alignment in LLMs focuses on preventing harmful or inappropriate outputs (Gehman et al., 2020; Yi et al., 2024). The most common method is Reinforcement Learning with Human Feedback (RLHF), where models are fine-tuned to align responses with ethical standards (Christiano et al., 2017; Ouyang et al., 2022). While effective, RLHF can fail against novel or adversarial inputs. Adversarial training, which exposes models to harmful inputs to increase resilience, is also widely used (Kumar et al., 2023; Cheng et al., 2024). However, these methods often degrade when models are quantized for edge deployment. Safety prompts (Röttger et al., 2024) offer additional control but are less robust post-quantization due to changes introduced in the process.

**Jailbreaking Attacks on LLMs.** Jailbreaking attacks exploit LLM vulnerabilities by manipulating inputs to bypass safety constraints and generate harmful outputs (Li et al., 2023; Mehrotra et al., 2023). Adversaries craft prompts to exploit model understanding, circumventing safety measures (Wei et al., 2024). While existing defenses (Robey et al., 2023; Röttger et al., 2024) are effective in full-precision models, they often fail in quantized models, where reduced capacity exacerbates vulnerabilities, making it easier for attackers to trigger unsafe behavior (Zhang et al., 2024).

**Quantization Conditioned Attacks.** Research has demonstrated that adversarial and backdoor attacks leverage the nuances of quantized weight distributions, leading to unpredictable model behavior (Gupta & Ajanthan, 2022; Li et al., 2024; Lechner et al., 2023). Recent work (Egashira et al., 2024) revealed how quantization can be exploited for vulnerabilities like code generation (He et al., 2024), over-refusal attacks, and content injection (Shu et al., 2023). These latent vulnerabilities make quantized models particularly prone to misalignment post-deployment, emphasizing the need for focused adversarial defense research (Wei et al., 2024). Inspired by Ma et al. (2023), which introduced quantized conditional backdoor attacks and highlighted the challenges of directly inducing attacks in quantized precision, our work adopts a two-stage paradigm to address these limitations.

## 8   CONCLUSION

This paper introduced the Quantization Conditioned Misalignment (Q-Misalign) Attack, which targets latent vulnerabilities in large language models (LLMs) that emerge only after model quantization. We demonstrated that these vulnerabilities can lead to significant safety risks, with quantized models becoming highly susceptible to jailbreak attacks, while maintaining robustness in their full-precision form. We also proposed Contrastive Task Vectors (CTV) to enhance the persistence of misalignment, showing that this method alleviates the effects of catastrophic forgetting during downstream fine-tuning. Our results highlight the limitations of current safety alignment techniques, such as RLHF and adversarial training, which fail to protect quantized models. This work underscores the need for quantization-aware safety strategies and opens avenues for developing robust defenses that ensure model safety across both full-precision and quantized environments.

ACKNOWLEDGMENTS

This research was supported by fundings from the Hong Kong RGC General Research Fund (152244/21E, 152169/22E, 152228/23E, 162161/24E), Research Impact Fund (No. R5011-23F, No. R5060-19), Collaborative Research Fund (No. C1042-23GF), NSFC/RGC Collaborative Research Scheme (No. CRS_HKUST602/24), Theme-based Research Scheme (No. T43-518/24-N), Areas of Excellence Scheme (No. AoE/E-601/22-R), and the InnoHK (HKGAI).

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
